# Barriers to reducing preoperative testing for low-risk surgical procedures: A qualitative assessment guided by the Theoretical Domains Framework

**Amanda Hall**[1]*, **Andrea Pike**[1], **Andrea Patey**[2], **Sameh Mortazhejri**[2], **Samantha Inwood**[1], **Shannon Ruzycki**[3], **Kyle Kirkham**[4,5], **Krista Mahoney**[6], **Jeremy Grimshaw**[2]

1 Primary Healthcare Research Unit, Memorial University, St. John's, Newfoundland and Labrador, Canada, 2 Centre for Implementation Research, Ottawa Hospital Research Institute, Ottawa, Ontario, Canada, 3 Department of Medicine, University of Calgary, Calgary, Alberta, Canada, 4 Department of Anesthesia and Pain Management, Toronto Western Hospital, University of Toronto, Toronto, Ontario, Canada, 5 Department of Anesthesia, Women's College Hospital, Toronto, Ontario, Canada, 6 Faculty of Medicine, Memorial University, St. John's, Newfoundland and Labrador, Canada

* amanda.hall@med.mun.ca

**Data Availability Statement:** Ethical restrictions prevent us from sharing the de-identified data set in a data repository. The data were collected from a

## Abstract

### Introduction

While numerous guidelines do not recommend preoperative tests for low risk patients undergoing low risk surgeries, they are often routinely performed. Canadian data suggests preoperative tests (e.g. ECGs and chest x-rays) preceded 17.9%-35.5% of low-risk procedures. Translating guidelines into clinical practice can be challenging and it is important to understand what is driving behaviour when developing interventions to change it.

### Aim

Thus, we completed a theory-based investigation of the perceived barriers and enablers to reducing unnecessary preoperative tests for low-risk surgical procedures in Newfoundland, Canada.

### Method

We used snowball sampling to recruit surgeons, anaesthesiologists, or preoperative clinic nurses. Interviews were conducted by two researchers using an interview guide with 31 questions based on the theoretical domains framework. Data was transcribed and coded into the 14 theoretical domains and then themes were identified for each domain.

### Results

We interviewed 17 surgeons, anaesthesiologists, or preoperative clinic nurses with 1 to 34 years' experience. Overall, while respondents agreed with the guidelines they described several factors, across seven relevant theoretical domains, that influence whether tests are

small group of healthcare providers working in our province making re-identification a risk. We do not have permission from our Health Research Ethics Authority to share the complete de-identified data set. However, all relevant data are included in the tables within the manuscript. Data requests may be sent to: Research Ethics Office Health Research Ethics Authority Suite 200, 95 Bonaventure Avenue St. John's, NL. A1B 2X5 Phone: 709-777-6974 Email: info@hrea.ca.

**Funding:** 1. JG, AH, KK, SR, APa, KM 2. JG, AH, KK, SR, APa, KM - De-implementing low value care: A research program of the Choosing Wisely Canada Implementation Research Network - grant number 398 527. 3. Canadian Institute of Health Research (CIHR) 4. https://cihr-irsc.gc.ca/e/193.html 5. No.

**Competing interests:** The authors have declared that no competing interests exist.

ordered. The most common included uncertainty about who is responsible for test ordering, inability to access patient records or to consult/communicate with colleagues about ordering decisions and worry about surgery delays/cancellation if tests are not ordered. Other factors included workplace norms that conflicted with guidelines and concerns about missing something serious or litigation. In terms of enablers, respondents believed that clear institutional guidelines including who is responsible for test ordering and information about the risk of missing something serious, supported by improved communication between those involved in the ordering process and periodic evaluation will reduce any unnecessary preoperative testing.

## Conclusion

These findings suggest that both health system and health provider factors need to be addressed in an intervention to reduce pre-operative testing.

## Introduction

Preoperative tests (e.g., chest x-rays, electrocardiograms (ECG), and baseline laboratory studies) are performed to provide additional information about high-risk patients (i.e., those with known risk factors identified via clinical history and physical examination) to help anaesthesiologists prepare them for surgery and improve perioperative outcomes [1–3]. In practice, however, preoperative testing is often performed using a routine testing strategy, defined by the Agency for Healthcare Research and Quality as tests conducted on all patients undergoing given procedures, regardless of patient history [4]. Without supporting evidence, many hospitals have chosen to implement routine testing as a "fail-safe," seemingly under the impression that more information (from more tests) will increase patient safety and decrease potential legal action resulting from adverse events [2].

The scientific foundation for the use of routine testing is very weak. A Cochrane review published in 2012 found that routine preoperative tests did not reduce the risk of intraoperative or postoperative adverse events when compared with selective or no testing for patients undergoing cataract surgery [5]. Similarly, another review on trials involving cataract surgery and various types of ambulatory procedures concluded that routine preoperative testing did not affect clinical management or reduce total perioperative complications, morbidity or mortality compared with no testing [4,6]. There are few studies on the effects of routine preoperative testing for other types of low-risk surgeries making it difficult to draw conclusions on the utility of routine preoperative testing within these settings [4,7].

Because routine testing lacks evidence to support its use, it is considered low-value care and current clinical practice guidelines and Choosing Wisely recommendations advise against routine preoperative testing for healthy adult patients undergoing low-risk surgical procedures [8–12]. These are procedures for which the combined surgical and patient characteristics predict a low-risk (<1% in American guidelines) of a major adverse cardiac event (such as death or myocardial infarction) [13,14]. In addition to its considerable cost [2,10,15], it is doubtful that any benefits realised using routine testing outweigh the drawbacks (e.g., extra testing, incidental findings that require additional costly investigations, surgical delays and patient stress/harm) [2]. Unfortunately, routine preoperative testing for low-risk surgeries persists. Between the fiscal years 2011/2012 and 2012/2013, preoperative tests (including ECGs, chest x-rays, stress tests, and transthoracic echocardiograms) preceded 17.9%-35.5% of low-risk procedures

across the Canadian provinces of Alberta, Saskatchewan, and Ontario [9]. Additionally, an Ontario study conducted using provincial data from 2008 to 2013 showed that ECGs and chest x-rays were conducted before 30.9% and 10.8% of procedures, respectively [16]. Similar figures have been reported internationally [10,17–19].

## Improving uptake of guidelines

Translating guidelines into clinical practice is challenging, and simple guideline dissemination is unlikely to change clinical practice behaviours that are influenced by a multitude of factors [20]. It is important to understand what is driving behaviour when developing interventions to change it [21,22]. Ideally, this process should be guided by a theoretical framework of established psychological theories of behaviour change [20–23]. The Theoretical Domains Framework (TDF) was developed by Michie et al. to identify factors influencing health professionals' implementation of evidence-based guidelines into practice [24–26]. The TDF simplifies 128 constructs from 33 behaviour change theories into 12 theoretical domains [26,27]. Each theoretical domain may be a determinant of the behaviour that requires change.

We are aware of only one Canadian study using a theory-based approach to examine healthcare providers' (HCPs) barriers to de-implementing routine preoperative tests for low-risk surgeries. [28] Patey et al found the most common barriers identified by anesthesiologists and surgeons practicing in Ontario, Canada were related to the lack of clarity regarding who was responsible for ordering the tests, perceived inability to cancel tests ordered by other HCPs, and tests being ordered and completed prior to the anaesthesiologist's examination of the patient [28]. While some of the drivers of behavior may be similar among HCPs across Canadian provinces, healthcare systems are provincially managed leading to contextual differences in each province. Indeed, even within provinces, practices can vary from hospital to hospital. We completed a TDF-guided investigation of our local context by conducting a series of semi-structured interviews with surgeons, anaesthesiologists, and preoperative clinic nurses in Newfoundland and Labrador (NL) to identify perceived barriers and enablers to reducing unnecessary preoperative tests for low-risk surgical procedures.

## Materials and methods

### Design

This was an exploratory, qualitative study conducted using semi-structured interviews to understand HCPs' barriers and enablers to de-implementing low-value preoperative testing. The protocol for this study was previously published [29]. This study was reported according to the Consolidated criteria for Reporting Qualitative research (COREQ) checklist [30].

### Participants

Participants were selected using a purposive snowball sampling strategy in order to ensure diverse perspectives and representation across the four regional health authorities (RHA) in NL, as well as a variety of surgical subspecialties. We identified a key informant within the largest RHA, Eastern Health, to provide a list of three potential participants as well as potential informants from the other three RHAs across the province. At the end of each interview, participants were asked to identify additional participants who may be interested in participating in this study.

Eligible participants were HCPs (surgeons, anaesthesiologists, or preoperative clinic nurses) practising in NL who order preoperative tests for patients undergoing surgery.

## Data collection

Potential participants were emailed by a researcher (KM or AH) to gauge their interest in participating in the study. Oral consent was obtained, and semi-structured interviews were conducted by two interviewers (AP and AH, both females, with graduate degrees in the health sciences, and employed as academic faculty and/or researchers). The interviewers were trained in qualitative methods and interview techniques, with over 15 years of experience. Participants learned about the interviewers at the start of the interview via verbal introduction; there was no relationship between the interviewers and participants established prior to the start of the study. Field notes were taken by a non-participant observer (RL or KM) during the interviews as a fail-safe measure in case of a recording failure) but were not used in the analysis. Interviews were conducted over the phone or in-person. All interviews were audio-recorded and transcribed verbatim. No repeat interviews were carried out. Transcripts were not returned to participants for comments and/or corrections and participant checking of findings was not performed.

## Interview guide

The behaviour of interest was ordering of preoperative tests (chest x-rays and ECGs) for healthy patients undergoing low-risk surgical procedures (knee arthroscopy, laparoscopic cholecystectomy, cataract removal, and similar types of surgeries). Healthy patients were defined as those without comorbidities or additional medical conditions that could complicate anaesthesia management and perioperative care [28]. The interview guide (adapted from Patey et al.'s study with surgeons and anaesthesiologists on preoperative testing in Ontario [28]) was developed using the TDF to elicit participant beliefs about their behaviour through the lens of each domain. It included 1–4 questions per domain (12 domains), for a total of 31 questions and prompts were provided in the interview guide to assist the interviewer in clarifying participants' responses if needed. No changes were made to the guide after pilot testing with our key informant. Please see the published study protocol for a copy of the interview guide [29].

## Data analysis

Following the method outlined in the TDF guide [26], coders read and reread transcripts to become familiar with the data. Coding began after six interviews were completed and transcribed. Using the TDF to generate a framework for content analysis, researchers analyzed the data deductively (assigning text to one or more domains) and inductively (identifying themes within each domain.

**Deductive analysis.** Two coders (SI, SM), trained in TDF coding by an expert in TDF and behavioral sciences (AMP), developed a codebook which served as a guide and reference for the coders to ensure accuracy and consistency. To develop the codebook, one interview was coded in NVivo (V.12, QSR International, Melbourne, Australia) in tandem with access to an expert coder (AMP) for review and correction. A second transcript was coded independently to validate the codebook and calculate interrater reliability using Fleiss's Kappa (κ) [31]. Domains with κ<0.8 (indicating less than 'substantial agreement' or 'excellent agreement' [32]) were reviewed for consensus. Any disagreements amongst the two researchers about coding were reviewed by the TDF expert (AMP). Once the coders were comfortable with their strategy, they continued to independently code the remaining transcripts, reviewing agreement every three interviews to ensure consistency. Again, any domains with κ<0.8 were reviewed and coded to consensus. Please see the published protocol for this study for a copy of the codebook [29].

**Inductive analysis.**    After interview responses were coded into TDF domains, themes were identified for each domain, phrased as belief statements about barriers or enablers to pre-operative test ordering. These were further examined to identify broader themes or patterns in the data. All belief statements and broad themes (with supporting quotes) were reviewed by the second coder and AMP.

## Identifying relevant domains

Relevant domains were identified through consensus discussion between the two coders (SI, SM), confirmed by AMP, and subsequently reviewed with the larger research team. Relevant domains are those for which sufficient data were coded to inform an intervention for behaviour change [33]. Three factors were considered to identify relevant domains: (1) reported strength of opinion that the beliefs influenced the behaviour, (2) presence of conflicting beliefs, and (3) frequency of the beliefs across interviews [26]. All factors were considered concurrently to establish domain importance.

## Ethics

Ethics approval was obtained from the Health Research Ethics Board in Newfoundland and Labrador (HREB #2018.190).

# Results

We interviewed seven surgeons, five anesthesiologists and five nurses from hospital (twelve) or hospital and community settings (five) who had practiced between 1 to 34 years (mean ± SD, 14.23 ± 9.33). Data saturation was achieved after 17 interviews. The interviews took between 13 to 76 minutes (mean ± SD, 38.97 ± 14.82). Initial interrater reliability for domains ranged from k = 0 to k = 1 (mean = 0.58 ± SD = 0.22). For Kappas below 0.8, disagreements were discussed in order to reach consensus for all domains.

## Preoperative assessment process

Participants reported different definitions of "low-risk surgeries," but generally, both patient characteristics (past medical history and physical exam) and type of the surgery were reported to play a role in defining a surgery as low-risk. Same-day surgeries (i.e., the patients who could go home after the surgeries) and those that did not require general anaesthesia were regarded as low-risk. Some participants considered patients of certain age groups (e.g., 50, 60, 70 years) as high-risk regardless of other factors.

There were inconsistencies among the participants when describing the process for preoperative assessment. Most often, patients were initially seen by surgeons. Based on their medical history, physical exam, and type of required anaesthesia, they were either referred to the pre-admission centres (PAC) or were booked directly for their surgery without any preoperative assessment. If patients were referred to a PAC, they would first undergo an assessment by PAC nurses (sometimes this would be done via telephone). The nurses reviewed patients' medical history, discussed the details of the surgery with them, and referred them to in-person visits by anaesthesiologists for further examination and tests (if necessary). However, in some centers, the anaesthesiologists' visits were only required if any red flags were highlighted by the nurses.

Although most participants reported that the tests were ordered at PACs by anaesthesiologists, tests were also ordered at surgeons' offices or by nurses during the telephone interviews. In some rare occasions, especially if patients were admitted as outpatients by surgeons, the anaesthesiologists would not see the patients until the moment of surgery.

## Key themes among relevant domains

Theoretical domains identified as relevant to preoperative test ordering were: beliefs about capabilities, beliefs about consequences, social/professional role and identity, motivation and goals, social influences, environmental context and resources, and behavioural regulation. Table 1 presents a summary of the belief statements and supporting quotes for each of the relevant domains.

Four overarching themes are described below

1. While HCPs may not *intend* to routinely order preoperative tests, in practice, a host of factors play a role in ordering decisions including use of automatic order sets, workplace norms, expectations about the needs and wants of other HCPs, and lack of access to information, technological tools, and human resources.

   Most participants stated that they did not intend to *routinely* order tests and were comfortable with the idea of not doing so. However, some believed that the tests are important to carry out to provide the clinical team with as much information about the patient (and potential adverse outcomes) as possible (motivations and goals, beliefs about consequences).

   Participants reported a variety of factors that impacted their ordering decisions, some of which were outside of their influence (e.g., institutional use of automatic order sets and

**Table 1. Summary of belief statements and sample quotes from anesthesiologist, surgeons and nurses assigned to the theoretical domains identified as relevant.**

| Domain | Themes | Belief statements | Sample quotes | Frequency |
|---|---|---|---|---|
| Social/professional role and identity | Inter-professional interactions/ conflicts | Anesthesiologists and surgeons have different goals and concerns relating to surgeries (Barrier). | 'There's always a little bit of friction between anesthesiology and surgery, and surgery wants to proceed, and anesthesia is happier to cancel. So, in my mind, it was always like let's do everything and that way there's no way they can cancel them.' (S2) 'They (surgeons) do a history and physical as well. It won't be in the same because they're looking for different things, so they won't be reviewing the patient on active cardiac and respiratory symptoms usually they're more just have a story surrounding their surgical problem and then a review of their systems and a past medical history of course.' (A14) | 8 |
| | | Abnormalities or patients I'm not sure about need to be reviewed by anesthesia or internal medicine (Barrier). | 'Any patient that we have any question about we ah, we have an understanding amongst our group to refer them either to internal medicine or anesthesia for preoperative workup if necessary, um before ah going ahead with the surgery.' (S16) 'If I do have any questions or concerns after that, I would put the patient for review, and I do ask the anesthetist at that time.' (N9) | 7 |
| | | Anesthesiology doesn't always see test results because they go to the surgeon, whose name the test is under (Barrier). | 'So, we (anesthesiologists) want to be informed, but laboratory doesn't inform us directly. Laboratory will inform the surgeon. . .We want to be informed if it's going to affect our ability to give the anesthetic. That doesn't always come through to us.' (A7) | 3 |
| | | Sometimes surgeons have to follow up on test results that are ordered by anesthesia (Barrier). | 'If they want tests done, it should be in their name because they're ordering the tests, right? It's not reasonable that there's a policy that says this test has to be done and it has to be done even if I don't think it does and then it gets put in my name and all of a sudden there's a test I didn't order that's in my name that I have to follow up on.' (S5) | 2 |
| | Professional expectations | Reviewing a CXR or ECG for low-risk patients/ surgeries is an expected part of my preoperative check (Barrier). | 'Is a review of chest x-ray or ECG an expected part of your pre-op check? Yes, it is right.' (N17) | 2 |
| | | I don't need to order ECG/CXR to do my job (Enabler). | 'If I do not order the tests, I believe I have done my job because I probably took a thorough history and physical.' (S16) 'If you're doing a pre-op evaluation or visit and you don't order an ECG or a chest x-ray, do you think you're doing your job? I'm doing my job based on the information that we have in front of us.' (N10) | 15 |
| | | If a CXR/ECG is ordered, I'll review it, even though it wasn't ordered by me (Enabler). | 'If it's ordered, that's a tricky one because if it's there on the chart, then yes, we should be reviewing this, if it's there.' (A8) | 5 |
| | | My professional role doesn't influence my decision to order (Enabler). | 'Is there anything in your professional role as a surgeon that influences your decision to order or not order certain tests for a patient having a low-risk surgery? I wouldn't say so.' (S4) | 5 |
| | | Ordering a CXR or ECG for all patients is not an expected part of my preoperative check (Enabler). | 'Is ordering a chest x-ray or ECG for every patient an expected part of the pre-op visit? Not anymore, thankfully.' (S2) | 12 |
| | | Reviewing a CXR or ECG for low-risk patients/ surgeries is not an expected part of my preoperative check (Enabler). | 'Is a review of a chest x-ray or echocardiogram an expected part of a pre-op check? I wouldn't say it's an expected part, no.' (S4) | 8 |
| | Test ordering | I'm not in charge of ordering (Barrier). | 'So, we don't make that decision, we gather the information and pass it along to the doctor (anesthesiologists).' (N17) 'I'm not the person ordering these tests, I'm kind of a step back from the process.' (A13) | 7 |
| | | Tests happen in surgeons' name without they ordering them personally (Barrier). | 'They're asking the patient to do certain things and I might not even know what they have asked. But I am supposed to know because it goes in my name, but then the decision-makers are the nurses and the anesthesiologists and the PAC clinic.' (S3) | 5 |
| | | Surgeon's secretaries order on behalf of surgeons (Barrier). | 'Again, it all happens just out of my hands, which I mean rightly or wrongly, a lot of this my secretary does and she's the one who has the guidelines in front of her and she has a—I don't know if I should say this on the recorder—but like a stamp with my signature on it and she knows what's what and I trust her. And so, she does a lot of that stuff that I never see.' (S2) | 2 |

*(Continued)*

**Table 1.** (Continued)

| Domain | Themes | Belief statements | Sample quotes | Frequency |
|---|---|---|---|---|
| Beliefs about capabilities | Comfort | I feel comfortable not ordering tests (Barrier). | 'I'm very familiar with the guidelines and the evidence, and it makes me very comfortable in not ordering these things.' (S1)<br>'You're comfortable proceeding without ordering those kinds of test for a low-risk patient? In a low-risk patient and low-risk procedure? Yes.' (A14) | 17 |
| | Ease of ordering | It is easy for me to order tests (Barrier). | 'How easy or difficult is it for you, personally, to order pre-op tests? It would be dead easy if I wanted to.' (S4)<br>'It's very easy to order the stuff and yeah all your colleagues like the nurses they'll help you get that done for sure whether it's drawing blood, or chest x-ray or EKG or stuff yeah it's not hard to do at all.' (A15) | 13 |
| | Ease of not ordering or cancelling | It's difficult to cancel tests once they've been ordered (Barrier). | 'Well if a physician orders something, I've got no right to go against them.' (N12)<br>'Trying to retract that once the patient disappears from your presence, it's very difficult, to be honest with you. Once you send them off to have something like that done, it tends to get done and you wouldn't have a chance to cancel it.' (S1) | 5 |
| | | Tests are done before I see the patient, I can't cancel or change orders (Barrier). | 'By the time we've seen them they've had their investigations done because they see the nurse, they're sent off to get their investigations done and then they come back and see us. So, it's almost always done by the time we see them. So those would be impossible to cancel because they're done.' (A15) | 2 |
| | | It is easy for me not to order tests (Enabler). | 'Personally, it is very easy for me not to order a test.' (S1)<br>'I guess it's fairly easy to order no tests.' (N10) | 8 |
| | | It isn't difficult to cancel tests once they've been ordered (Enabler). | 'Is it difficult or easy for you to cancel tests? No, it's not difficult. I mean that doesn't happen very often.' (N11) | 4 |
| Beliefs about consequences | Consequences | By not testing, you may miss something (Barrier). | 'We feel it's very important to have them done just so that everybody's aware of any situations that may arise with the patient while they're here.' (N9)<br>'The negative is that you might miss something that is maybe subclinical. So, you don't catch it on your physical exam, you don't catch it on your history but there is something on the chest x-ray or something on the EKG, something in the bloodwork that you aren't aware of.' (S6) | 5 |
| | | Sometimes surgery might get cancelled or delayed without testing (Barrier). | 'The only potential negative that I can see is the patient who is unnecessarily delayed or canceled because somebody else is not aware of or refuses to adhere to the guidelines.' (S1)<br>'I know some of my colleagues are far more lax about caring about certain tests or investigations but there's some that are like why didn't you order this I'm going to cancel this case until that's ordered or delay it until I get my EKG or my repeat CBC or whatever.' (A15) | 5 |
| | | By not testing, you can avoid inconveniences to the patients, false positives and unnecessary follow ups (Enabler). | 'So, positives, you save the system money and the patient time as well as the system time. You have fewer false positives, which can lead to more tests and more confusion.' (S6)<br>'You know you're fine. You've always been fine but what if they find something? That stress is unnecessary stress. Then there's the actual procedure which can be uncomfortable and then if it's not necessary, why do it?' (A7) | 9 |
| | | Medically speaking, there is no downside to not ordering preoperative tests (Enabler). | 'I think, medically speaking, there is no downside to not ordering these tests because I think they are useless.' (S1)<br>'What do you think would happen if you don't order pre-op tests, say, to patients? I don't think there are any negative consequences.' (N11) | 9 |
| | | Not ordering is more efficient, cost-effective and avoids wasting of resources (Enabler). | 'Long-term, it (not ordering tests) would decrease resource use and resource waste and probably allow easier access to those resources.' (S4)<br>'We're ordering tests that we don't need alright. However, once they have arrived you know I can't do anything about the tests they've already done and it's a complete and utter waste of money.' (A13) | 12 |
| | | Not ordering reduces wait times for surgeries (Enabler). | 'It (not ordering tests) would get people probably to the OR a little faster because they wouldn't be waiting for tests.' (S4) | 3 |
| | | Not ordering tests means less paperwork (Enabler). | 'I think the positive thing if we order less things would be just that we have less things coming through our inbox, which is a wonderful thing, especially now with the EMR. Oh my God, it's full all the time of crap that I don't want to see.' (S2) | 4 |
| | | The results of these tests are unlikely to change my practice (Enabler). | 'I have to follow up on it even though it was not a test I ordered. And that is absolutely infuriating. Mostly because it may not affect the surgery and the clinic that ordered it to help with the surgery isn't actually following up on their own test results, which is inappropriate.' (S5)<br>'Testing in general doesn't add much to anesthesia I don't think if you can get a decent history and physical from them there's very little else information that you're getting so. Predicted values are low.' (A15) | 10 |
| | Reasons for ordering | Tests are ordered because of fear of cancelling or delaying the surgery by anesthesia (Barrier). | 'The only reason a surgeon would order any of these tests is a fear that the patient would be canceled by anesthesia if these tests were not ordered, or a case would be delayed in some fashion.' (S1)<br>'But generally speaking, is driven by the surgeon and to some extent what they order is try and keep the anesthetist happy, so they don't get a case cancelled.' (A13) | 7 |
| | | If I'm concerned and not sure, I'll order (Barrier). | 'If I was truly worried I would just order it right and I think I'm you know confident enough to that and some of my colleagues I think for the majority would be fine just saying so but I'm just going to order so.' (A15) | 4 |
| | | I sometimes order tests to avoid conflicts with other colleagues (Barrier). | 'When it comes to the patient getting affected because I am in conflict with another provider, a certain test which I don't think it is necessary and the other person thinks vice versa, then I don't have much say in cancelling it and I will go ahead, because at the end of the day, I would have to think that will benefit the patient.' (S3) | 3 |
| | | Sometimes, tests are ordered to avoid legal issues (Barrier). | 'If I don't order this and I miss it, then I could be taken to court and sued. So that's what has guided a lot of investigations over the years. I'm doing this to make sure. I'm doing this to cover my butt.' (A8) | 2 |
| | Attitudes | Positive consequences of not testing outweigh negatives (Enabler). | 'Overall would you say that the positive consequences of not ordering um outweigh the negatives or the other way around? No, I for sure would say that they outweigh the negatives.' (S16) | 9 |
| | | Lots of testing we do is unnecessary (Enabler). | 'In order to not miss a few percentages of patients, 90 per cent of patient are having unnecessary investigations.' (S3) | 3 |
| | | There is no benefit to routine testing for all patients (Enabler). | 'The idea of blanket ordering isn't well established or isn't a great practice.' (A14) | 3 |

(Continued)

**Table 1.** (*Continued*)

| Domain | Themes | Belief statements | Sample quotes | Frequency |
|---|---|---|---|---|
| Social influences | Impact of other health professionals | Other HCPs impact me to order more tests (Barrier). | 'So, if I have an anesthetist who indiscreetly asks for an x-ray or an EKG, I will have to go with his or her decision and I don't have a say in it.' (S3)<br>'We'll do things that may not be required on the grid, but the physician has ordered it. We always follow the physicians' orders first and foremost.' (N9)<br>'The other side of it is that surgeons want these investigations done and then when I'm in pre-admission, I don't agree with that so sometimes I struggle... then we're there going, well you don't need all this so the majority of times I say no. But then there's sometimes that you say okay, go ahead if that's what you want, even if you don't agree.' (A8) | 12 |
| | | Fear of anesthesiologists cancelling impacts my decision to order (Barrier). | 'The only other member of the team that would cause me to rethink it would be my anesthesiologist. Again, it would be in a situation where you would worry that they would delay or cancel a procedure for one of these tests.' (S1) | 6 |
| | | I don't cancel tests ordered by others (Barrier). | 'I can't cancel another test that is ordered by somebody else for my patient because I'll ask for their help, anesthesiology.' (S3)<br>'If a surgeon has ordered stuff it's really hard for me to cancel their investigations because there maybe some surgical risk or something that I'm not aware of I'm not a surgeon that they may have actually wanted and most of the time your secretary just ticks these boxes, this is junk you don't need it but sometimes they may have actually put some thought into it and that's where I'd be hesitant to cancel what they've actually ordered.' (A15) | 6 |
| | | My decision is influenced by the norm of where I work (Barrier). | 'I guess the biggest thing now is, you know, having the, I guess, confidence not to order and having the backup of your colleagues that if I didn't order something and something happened positive, that's not going to come back to reflect negatively on me. So, I have to have agreement with all my colleagues or people there that what I'm doing is appropriate and acceptable.' (A8) | 3 |
| | | My testing decision is sometimes influenced by avoiding inter-professional conflict (Barrier). | 'But I also feel that I'm the lowest voice on the rung of the ladder that somebody else is going to trump me and I'm perfectly happy to say okay fine, do whatever you want, I don't care. But yeah, I'm not going to fight them. Put it that way.' (S2) | 4 |
| | | Sometimes I order tests because I anticipate that's what anesthesiologists would require (Barrier). | 'So, when I'm in pre-admission, we have a list and I know which anesthesiologists are going to be giving the anesthetic to the patient and I do take them into account, because I may not want an EKG and I'm thinking they're going to want it or, you know, they're going to delay surgery down there and have one. So, it does influence what I order on who is going to do the anesthetic.' (A8) | 3 |
| | | My colleagues and I have the same opinions about test ordering (Enabler). | 'There are three nurses in our unit that we do ask each other opinions on things when it comes to patients and we usually are pretty agreeable with things.' (N11)<br>'Do you colleagues generally agree with you on this issue? I think the majority do. I think the majority don't test unnecessarily.' (A7) | 16 |
| | | Others don't impact my ordering decision (Enabler). | 'Would any other team members, say, influence whether or not you order certain tests for a patient? So that might be other nursing staff or clinicians or-? No, not really.' (N11) | 6 |
| | Impact of patients | Patients don't influence my decision to test (Enabler). | 'Do patients ever influence whether or not you order certain tests for a pre-op evaluation? Their clinical conditions might. Their opinion of whether they should have a test wouldn't.' (S4) | 13 |
| Motivation and goals | Motivation | There is no incentive or disincentive for me to reduce testing (Barrier). | 'Are there any incentives or disincentives for you to reduce the number of pre-op tests you order when evaluating patients? There are no incentives or disincentives for me personally, no.' (S6) | 5 |
| | | I want to practice good medicine (Enabler). | 'The best practices are a reasonable thing and the reason why they do research on all these different things is to try and come up with what's the best approach. So, we try and follow that as best we can, given the limitation of the organization.' (A13) | 6 |
| | Importance of ordering | It is important to me to test (Barrier). | 'We feel it's very important to have them done just so that everybody's aware of any situations that may arise with the patient while they're here.' (N9) | 3 |
| | | I don't intend to routinely order tests (Enabler). | 'For patients having low-risk surgery, do you plan to routinely order ECG or chest x-rays? Is it something you feel you need to do? No.' (A8) | 10 |
| | | I don't feel I need to routinely order tests (Enabler). | 'For patients having a low-risk surgery do you plan to routinely order ECGs and chest x-rays. Is that something you feel you need to do? No, it's not anything I feel I need to do.' (S1) | 5 |
| | | It is not important to me to run these tests (Enabler). | 'How important is it to you to perform pre-op tests for a patient having a low-risk surgical procedure? Not important for me, personally at all.' (S4) | 6 |
| | | It's important for me to do the tests if only they are necessary (Enabler). | 'But if they're–a good reason for having a test done then we wait for the result and order the test.' (A13)<br>'I think if a test is really needed, I would still order it.' (S2) | 7 |
| Environmental context and resources | Resources | Automatic order sets and pre-defined templates can trigger ordering (Barrier). | 'We've tried to remove order sets within the hospital, but I can tell you they still exist and they still float around and some of them will have the special needs CBC, da, da, duh and they'll have a whole string of tests that theoretically you're supposed to think about before you tick them.' (A7) | 5 |
| | | Difficulty in accessing other specialists and consulting with them causes unnecessary test ordering (Barrier). | 'Now I've found something in pre-admission, we don't have any setup, especially at [hospital], I want another opinion or I want an internist or I want a cardiologist to look at it. . .It's very difficult to get someone to see them so therefore, that will turn me around and I will probably investigate more. . .then I may tend to order more tests on them, than probably necessary that if the specialist in that area is seeing anybody, say okay, you don't have to do this.' (A8) | 1 |
| | | Lack of easy access to technology and online tools impact ordering (Barrier). | 'If someone is being admitted for say surgery there's certain boxes they're going to tick. They're not going to go looking at previous, what you had done before because there's a laziness element. So, if the availability of the information or the previous information was more readily available then they're more likely to do it.' (A13) | 5 |
| | | Lack of staff affects preoperative assessment (Barrier). | 'But maybe I know that this surgeon isn't as skilled as someone else is more likely to bleed or maybe I know that this procedure is being done in a remote site and I won't have assistance from some of my colleagues so I want to have my ducks in a row a little bit better.' (A15) | 2 |
| | | The guidelines are unclear about what is considered low-risk (Barrier). | 'I think where things start to get grey is in the ASA 2, ASA3 zone, where they've got some disease, they're probably fine. Do you need to test them? Do you not need to test them? That's where we can have academic masturbation about what needs to be done and what doesn't need to be done.' (A7) | 1 |
| | | Some hospitals will repeat testing done at another hospital (Barrier). | 'If we do a test here bloodwork, x-ray, CAT Scan whatever if they go to [city] to have something done they will repeat the tests because they don't trust–they think somehow their machine is better. . .They will repeat the test.' (A13) | 1 |
| | | There are no environmental constraints that impact testing or ordering (Enabler). | "I mean, in a hospital setting there are really no barriers because the facilities are right there. There aren't physical reasons or barriers not to order them.' (S1)<br>'Everything is quite feasible to have done within the period of time that's required so that wouldn't influence.' (N9) | 13 |
| | | Forms allow us to check 'no testing needed' (Enabler). | 'There is a box that you could check there right, no preoperative testing needed.' (N17) | 2 |
| | Time | Tests are ordered before I see the patient (Barrier). | 'They have fit healthy patients that are coming for a low-risk surgery to do peripheral surgery on their arm but because they went through ER, they've had every test under the sun because they came through ER.' (A7) | 3 |
| | | Time constraints can rush the assessment process (Barrier). | 'I don't know if it's expertise as much as time thing. You know, you need 15 to 30 minutes to do a detailed preoperative assessment and I don't have time to do that in my clinic because I've got other people to see.' (S5) | 5 |

(*Continued*)

**Table 1.** (Continued)

| Domain | Themes | Belief statements | Sample quotes | Frequency |
|---|---|---|---|---|
| Behavioural regulation | Guidelines and policies | Clear guidelines, educating everyone on the evidence and having everyone on the same page will facilitate appropriate testing. | I think you would need a very strong framework of the team being on the same page. I think that's the main bit of it is that you would need to have nursing and anesthesia and the surgeons all adhering to the same set of guidelines and a very strong set of guidelines in terms of this is what you need: x, y, z for all the different parameters that can exist.' (S2) There are always guidelines needed. There are always personal opinions. There are always grey areas between anesthetists, physicians and nurses, and unfortunately, you know, you always want to cover your own butt, no matter what discipline you come from. So, it's great to have something that is accepted,. . .So it is always nice to sit down and certainly talk it over and make sure everybody's on the same page.' (N9) | 15 |
| | | Institutional policies/guidelines and endorsement by regulatory bodies can help with proper testing. | As long as I have some backup to it. I think if it was me all by myself, I'd tend to do the same old thing: order everything and that way I have all the boxes ticked and I can be sure we're going to move on. But as long as there is somebody else behind me saying oh no, we don't need this. I'm like oh yeah. Of course, we don't need it. . . As long as I have a guideline or something to back me up and it's not just me standing on a soapbox going no.' (S2) Sometimes the tests are motivated to do because people are worried about litigation you know. If you did the test and it was an adverse outcome or you didn't do a test, then are you kind of somehow responsible? How do you solve that problem? Well again you come back to your guidelines obviously sanctioned by a for example a national body which would be reasonable.' (A13) | 8 |
| | Inter-professional relations | Changing the accountability of test ordering would improve testing. | It would also be ideal if you had a pre-admission doctor who could order all the tests in their name, deal with all the results and then quarterback everything as to where everybody has to go, rather than putting my name on it and I get this thing going I don't know now. I don't know what to do with this.' (S2) | 3 |
| | | Communication with anesthesia can improve test ordering. | Sometimes maybe if our anesthetist was in our department where it is a little bit more anesthesia driven than surgeon driven, I don't know if we had an anesthetist directly in our department to ask well, you know, do you think that based on what we've obtained, do you feel that I should order this or not.' (N11) | 4 |
| | | Having family physicians do baseline testing would reduce preoperative testing. | Get the family doctors involved more and get the family doctors to do their baselines so we don't have to do that later or the surgeon doesn't have to do it. So, I think initially, it should start at the family doctor's office. That would make a big difference in what we're doing.' (A8) | 2 |
| | | Knowing that another colleague (anesthetists, nurses) will check my charts, makes me more comfortable not ordering. | Actually, I think what helps us as well is when anesthetists come and review the charts and if they feel that something needs to be ordered, they will probably say can we order a repeat whatever in the morning of surgery or something.' (N10) | 3 |
| | Feedback | Audit and feedback on accuracy of preoperative assessment and feedback on the impacts of not ordering helps/would help with appropriate test ordering. | I would think that individual specialities, they have to get together and should have the stats, some metrics as to what happened in the last six month, how many low-risk surgeries you did and what are all the investigations that have been ordered and do you really need them? And do you think you can revise them and check whether you could have ordered less or more? And what was the outcome when you didn't do it and what was the outcome that when you did everything and you wasted so much money and even indicating to every physician or whatever healthcare professional who is ordering the test as to how much each test costs so that they are fiscally aware as well, right?' (S3) | 5 |
| | Preoperative assessment | A thorough, accurate clinical history and physical exam improve appropriateness of testing. | You have to check the patients' chart and here the patients would have had some tests within the last six months, so you have to correlate if they are available, existing information on the same patient and the current health status.' (S3) | 8 |
| | | Detailed documentation (patients' symptoms, reasons for testing/not testing) can prevent further testing. | Any nuances of symptoms then, we feel we have to investigate before going ahead. But if this was documented beforehand, then I wouldn't worry about it and I wouldn't go on any investigations in pre-admission so it wouldn't make a difference. . .some of those processes, probably if at the surgeon's time of referral and they're doing their assessment, if at least the assessment was documented completely and fully with the results, that might be better rather than just there's no ordering of tests necessary.' (A8) | 1 |
| | | Face-to-face pre-admission assessments, instead of phone interviews, can help in decreasing testing. | I just think that sometimes maybe things wouldn't get ordered if I was doing it face-to-face. Maybe more things would be getting ordered if I did it face to face, but I don't know if that will ever come to that.' (N11) | 1 |
| | | Online tools (e.g. Meditech or HEALTHe NL) allow accessing the information which patients might not remember, and this will prevent unnecessary testing. | So, you know, we look up through our Meditech system or through the HEALTHe NL of visits that patients have made. Like we may ask a question "have you ever seen a cardiologist?" And some patients will say no, I've never seen anybody. And then you say well, you've seen such and such, maybe five years ago. Oh yeah, that's right. But sometimes they don't divulge that information.' (N11) | 2 |
| | | Pop-up questions when ordering a test (asking if patients' symptoms are new), could improve appropriate testing. | Maybe if you have like surgical admission orders check this and if they have to—if they want to check EKG or chest x-ray they have to ask the question are they have previous or new symptoms and then actually have to go ask the patient rather than just check it.' (A14) | 1 |
| | | Removing or changing order sets can help in reducing test ordering. | And order sets and pre-existing written orders, routine—the bane of my life, the word "routine". There should be no such thing as a routine preoperative order because that's the problem is that people have become automated in it. And in fact, we changed some of our order sets to say that testing as per our policy and as per pre-anesthetic clinic nursing assessment.' (A7) | 2 |
| | Timing and resources | If testing could be done closer to surgery, that would help us and patients (avoid follow-ups, travelling). | First of all, the screening questionnaire should not happen less than a week from the time of surgery so that you know whether there are any ongoing issues closer to the date of surgery.' (S3) | 3 |
| | | More support staff and/or resources would improve ordering. | So in an ideal world, I could have a GP or a nurse practitioner or a physician to do a complete history and physical exam instead of my like less thorough one, as well as completely look through all of their past medical history, which is essentially the role of a resident as well, but otherwise, like I'm essentially doing that myself.' (S6) | 3 |

pre-defined templates for preoperative evaluations (environmental context and resources) or tests ordered by other health care providers that they were unable to cancel (beliefs about capabilities). Other factors that influence ordering decisions include perceived needs/ requirements of other HCPs, workplace norms, and avoiding inter-personal conflicts (social influences). Fear of having procedures delayed or cancelled by anaesthesiologists if tests were not completed and fear of legal issues if patients were to experience an adverse outcome were also reported to influence ordering decisions (beliefs about consequences). Finally, while participants indicated a lack of environmental constraints to ordering, they

did highlight environmental factors that make it more difficult for them to *avoid* routine ordering including a) a lack of access to technology and/or online tools (which makes it difficult for them to check past medical history of their patients and the results of any tests they have undergone), b) lack of staff and difficulty in accessing other specialists for consults (environmental context and resources).

2. HCPs believe that the positive effects of not ordering outweigh the negative and don't think the tests are likely to influence their practice but some still consider them a fail-safe–to ensure procedures move ahead as scheduled and that important clinical findings are not missed.
   Most participants agreed the benefits of not ordering tests outweighed any potential risks (beliefs about consequences). Reported positive effects included reduced patient inconvenience, fewer false positives and unnecessary follow-up tests, less paperwork, reduced wait time for surgeries, and higher efficiency and cost-effectiveness (beliefs about consequences). They also stated that the results of the tests were unlikely to change their practice. Nevertheless, some participants noted that failing to order tests may result in surgery delays or cancellations or missing important clinical findings (beliefs about consequences).

3. There is a lack of clarity about who is responsible for deciding whether or not to order preoperative tests, though most agree that chest x-rays and ECGs are not required for low-risk patients.
   There was significant confusion and conflicting comments among the participants with regards to responsibility for preoperative test-ordering (social/professional role and identity). The surgeons we interviewed reported that they have a different set of concerns and goals for surgery (compared to anaesthesiologists) and may therefore want to order different tests to prepare for surgery. They also reported that if they were unsure or concerned about a specific case, they would refer it to their anesthesiologist or internist colleagues for consultation (social influences). While both groups agreed that chest x-rays and ECGs for low-risk patients undergoing surgeries was not an expected part of preoperative assessments, it was not clear who was responsible for deciding whether or not to order these tests. Rather, we found that preoperative test-ordering (meant to support anesthesia management) was not at the sole discretion of the PAC or anaesthesiologists (social professional role and identity). Sometimes surgeons order these tests in an attempt to ward off potential cancellations or delays. In addition, participants reported that tests were sometimes ordered under surgeons' names by other HCPs without the surgeons' awareness. Consequently, the HCP ordering the test could not check the test results because they were sent to the surgeons who then became responsible for following up on the tests they had not ordered. A few participants also reported that sometimes surgeons' administrative staff would order tests on their behalf using order sets for all patients regardless of patients or surgery characteristics.

4. HCPs believe that implementing clearer guidelines, enhancing understanding of current evidence, and building consensus on a preoperative testing strategy, supported by improved communication and access to resources, and periodic evaluation will reduce unnecessary preoperative testing.
   Participants emphasised the importance of clearer guidelines, educating stakeholders on the current evidence, and ensuring that there is consensus among stakeholders regarding preoperative testing strategy to reduce routine preoperative testing (behavioral regulation). They also explained that endorsement of policies and guidelines by regulatory bodies could help reduce inappropriate test ordering. Several strategies were mentioned to help reduce

test ordering: performing a thorough clinical history and physical exam, detailed documentation of patients' symptoms and reasons for testing/not testing, easy access to online tools (e.g., Meditech or HEALTHe NL), enabling pop-up questions when ordering a test (e.g., asking if patients' symptoms were new), and removing or changing order sets (behavioral regulation). Participants also suggested that audit and feedback on appropriateness of preoperative assessments and reminders about the benefits of not ordering could reduce test ordering. In terms of inter-professional relations, changing the accountability of test ordering and better communication with anaesthesiologists were highlighted.

### Themes among domains identified as not relevant

Table 2 presents a summary of the belief statements and supporting quotes for each of the four domains that were not considered relevant to preoperative test ordering: knowledge, skills, memory and decision making, and emotion. Most participants were aware of the preoperative guidelines and believed them to be evidence-based and trustworthy. A small number of participants were not sure about the details of the guidelines (knowledge). Almost all participants believed that basic clinical skills were enough for performing a history/physical exam and deciding whether tests were needed and that experience could improve the skills needed for preoperative assessments. They also felt that good interpersonal and communication skills with patients were important (skills).

The decision to not order tests was an easy decision to make by most participants. However, as mentioned previously, some participants recognised that automaticity and environmental factors influenced their test ordering decisions and explained that they were mandated to automatically order tests according to policies set by hospitals, for example, if patients were a certain age (memory and decision processes). Almost all participants mentioned that not ordering tests did not worry them (emotion).

## Discussion

This study used a theory-informed analysis to help understand the factors influencing low-value preoperative test-ordering by surgeons, anaesthesiologists, or preoperative clinic nurses in NL, Canada. We found that while HCPs may not *intend* to routinely order preoperative, in practice, a host of factors drive ordering decisions. Further, most participants believe that the benefits of not ordering outweigh potential risks, yet still order these tests in an attempt to ensure procedures move ahead as scheduled and that important clinical findings are not missed. Participants also reported a lack of clarity about who is responsible for deciding whether to order preoperative tests, though most agree that chest x-rays and ECGs are not required for low-risk patients. Respondents believe that implementing clear guidelines (at the institutional and/or health system level), enhancing understanding of current evidence, and building consensus on a preoperative testing strategy, supported by improved communication and access to resources, and periodic evaluation will reduce unnecessary preoperative testing.

### Findings in relation to previous research

We found only three studies investigating why routine testing practices persist despite guidelines that advise otherwise. Two were very similar to our study parameters including interviews with surgeons, anesthesiologists, and/or nurse administrators (conducted in the US and Canada) [28] [34]. The third interviewed general surgeons involved in preoperative clinics in New Zealand [35].

Consistent with these studies, participants in our study agreed with guidelines recommending against low-value, routine preoperative test ordering and did not believe that these tests

**Table 2. Summary of belief statements and sample quotes from anesthesiologist, surgeons and nurses assigned to the theoretical domains identified as irrelevant.**

| Domain | Themes | Belief statements | Sample quotes | Frequency |
|---|---|---|---|---|
| Knowledge | Knowledge of guidelines | I am aware of the guidelines. | 'I guess when the Choosing Wisely stuff comes across or whatever and I review it, maybe I'm totally off base but I think I try to keep relatively current with the evidence.' (S4)<br>'Are you aware of any guidelines about preoperative testing?<br>Yes. In general, when we go to these various annual meetings and so on yes, they do talk about these things and what we should be doing.' (A13) | 14 |
| | | I know what the guidelines say. | 'Do you know what they say about the pre-op testing? They stratify risk and they give, what test should be done preoperatively on what types of patients.' (S1)<br>'If someone is young and healthy without any medications, without any sort of age over certain limits you wouldn't necessarily investigate them.' (A15) | 8 |
| | | I'm not sure about the details of the guidelines. | 'I have read them in the past but not in the recent five years' time and I know through our pre-anesthetic clinic and PAC they follow Canadian guidelines. I haven't gone through them myself, no.' (S3) | 5 |
| | Knowledge of evidence | I believe the guidelines are evidence based. | 'Would you say they (guidelines)'re evidence based? Well I think they're evidence–well they're certainly evidence based.' (A15) | 10 |
| | | I trust the guidelines. | 'What are your thoughts about those pre-op testing guidelines in general? Do you trust what they say and those types of things? Yeah, I think they're great.' (N9) | 8 |
| Skills | Required level of skills | One needs basic clinical skills to do a history/ physical exam and decide to order/not order the tests. | 'Specifically ordering the tests you'd have to just be able to understand what the red flags are of cardiac disease and respiratory disease really.' (A14)<br>'If you can draw a check in a box, then that's really all the skill you need.' (S5) | 16 |
| | | Experience improves clinical skills to conduct an accurate assessment. | 'You have ways of getting information out of patients and the more patients you see the more experience you get, the better you should be at it.' (S1)<br>'I think coming out initially and inexperienced people, you're afraid to miss something so therefore you over-investigate, over-evaluate. And I think the time and expertise and that you see so many people, you get a better handle on them and you don't investigate as much'. (A8) | 9 |
| | Required skill sets | A good skill set is to consider both patients' characteristics and the characteristics of the surgery/anesthesiology. | 'You need to be able to do a comprehensive history and physical on the patient. You need to have a good understanding of what the surgery entails. Umm, and be able to specifically ask questions ah, on things that could affect the surgery. To find out if the patient has a possible comorbidity or a documented comorbidity that would hurt the outcome of the surgery.' (S16) | 5 |
| | | Good interpersonal and communication skills with patients are important to conducting an accurate assessment. | 'I mean, you need to be able to complete an accurate history, which means you need to get the information out of the patient, and some people are easier to get that out of than others are. You have to be able to deal with the difficult communicator.' (S1) | 4 |
| Emotion | Concerns | Not ordering tests doesn't worry me. | 'Does not ordering pre-op tests for a patient having a low-risk surgery evoke worry or concern in you? Not at all.' (S4) | 15 |
| Memory and decision making | Ease of decision | The decision to not order is an easy decision to make. | 'Is not ordering the tests typically an easy or difficult decision to make? It's an easy decision.' (A8) | 11 |
| | Decision making | According to hospital policy, I have to automatically order if patients fall under certain criteria (e.g. certain age). | 'If a patient falls within certain criteria, they automatically get certain testing or certain referrals.' (S16)<br>I would say at this point in my clinical practice is that I routinely order EKG's for people over 50 regardless of their clinical conditions.' (S4) | 9 |
| | | My decision to order/not order tests is based on patients' history, exam and medical condition. | 'If I haven't had anything on history or physical that suggests there's going to be a problem, then I'm not going to order it (S5)'.<br>'I'm not doing an EKG, a chest x-ray if the patient is asymptomatic or clinically stable.' (A8) | 16 |
| | | My default decision is to not order. | 'My default would be not to order the test and what happens with me is, I need a reason to order the test.' (S1) | 4 |

help or influence their practice [28,34]. Rather, they believe not ordering tests routinely can benefit patients (for example, by reducing patient discomfort and inconvenience) and the health system (for example, by reducing costs and improving efficiency) [28,34]. Additional practitioner-level barriers also noted in this study and the wider literature include the fear of missing "something serious" that could lead to adverse patient outcomes [34,35], medicolegal concerns, [34], worry that procedures will be cancelled or delayed if preoperative tests are not carried out [28,34], and beliefs about the preoperative testing needs and expectations of other specialties [28,34]. Further complicating the issue is confusion about professional responsibility (i.e., lack of clarity on who is responsible for ordering preoperative tests)–an issue also noted by Patey et al [28].

Our findings of systems-level drivers of routine testing were also reflected in the available literature. Brown & Brown (2011) and Raina et al. (2019) also found that practice traditions or workplace norms (e.g., routine testing strategies) were important barriers to reducing preoperative testing [34,35]. Additionally, Raina et al. (2019) also found that poor systems of communication between the HCPs and the departments involved, and a lack of staff or HCP time available for consults [35] are important barriers to change. These barriers, particularly poor communication between professions, is perhaps not surprising or unique to preoperative testing. Communication failures among teams of HCPs have been observed in many areas of healthcare resulting in redundancy, treatment errors, or even patient harms [36–42].

In contrast to other studies in this area [34,35], participants in this study were aware of preoperative testing guidelines and believed them to be evidence-based and trustworthy. This replicates findings from a Canadian study [28] which also didn't find that knowledge was a barrier to reducing preoperative testing and which had very similar study parameters and used a very similar interview guide. Guidelines recommending against routine preoperative testing strategies have been published for many years [43] and in recent years, Choosing Wisely has been actively campaigning on this issue which may partially explain why knowledge doesn't appear to be lacking among the participants we interviewed. Also, the questions about knowledge in the US [34] and the NZ [35] studies were focused on familiarity with the evidence around the benefits of preoperative testing, which guidelines were used in decision-making, and, in the NZ study, knowledge of the Choosing Wisely campaign. Globally, then, its not clear that knowledge is a major barrier but it does not appear to be the case in Canada.

Our participants also described additional contextual factors that drive routine testing including automatic order sets or pre-defined templates for preoperative evaluations, as well as lack of access to technology allowing for easy access to patient information. This issue was not initially included in our interview guide but came up in an early interview and was probed in subsequent interviews. This may be an issue relevant only in the local context, or it may not have surfaced previously simply because it was not asked about.

Finally, we found the desire to avoid interpersonal conflicts with colleagues also influenced physicians to order routinely. We believe this finding is related to other common barriers such as HCPs' beliefs about the needs and expectations of other HCPs and confusion about which HCPs are responsible for ordering. We were aware that previous studies had noted these types of issues so we knew to probe them if they came up in our own interviews resulting in a more nuanced understanding of this issue.

## Strengths and limitations

As recommended by numerous health and research organizations, (e.g., the National Institute for Healthcare Excellence, the MRC, Health Canada, Canadian Institutes of Health Research, and the Quality Enhancement Research Initiative) [44–48], this study used a theory-guided

investigation of the determinants of routine preoperative testing for low-risk surgeries. Using the Atkins *et al* guide [26] on how to apply the TDF to guide our assessment and analysis allowed us to produce results that can be used to develop a theory-informed intervention best-suited to tackle known barriers to reducing routine testing. Additionally, we applied rigorous data collection and analysis methods (including a TDF-based question guide, extensive coder training, double coding of all transcripts, oversight of analysis by a professional expert in the TDF, and review of results interpretations with clinical experts and the investigative team. interviews until data saturation, double coding, coding to consensus, interpretation checked by experts and reviewed with team). We also published the study protocol which was reviewed by content experts in the TDF, qualitative research, and preoperative testing and used the Consolidated Criteria for Reporting Qualitative Research (COREQ) 32-item checklist to guide our methods and reporting [30].

Despite this, our results are limited in several ways. We were unable to interview participants working at all sites offering low-risk surgical procedures in the province as originally planned. However, we were able to include participants from all but one regional health authority. In addition, although we reached data saturation, it is possible that we did so prematurely by not interviewing participants with sufficient diversity to allow for more variety in responses. For example, while we were able to speak with HCPs in three of four provincial health regions, we did not interview providers working in Labrador who may face different barriers to reducing routine preoperative testing. Finally, we didn't assess the test-ordering practices of our sample to engage similar numbers of participants that order these tests at different rates; nor did we actively seek participants in equal numbers who had differing views on routine preoperative test-ordering.

## Implications for research and practice

While HCPs interviewed for this study were aware of when to order preoperative tests according to guidelines, there are both system and practitioner-level barriers that continue to influence their practice. Many of these barriers occurred as a result of the complex nature of preoperative care and a lack of communication among providers suggesting that implementation is likely to be challenging. This challenge in implementing guidelines is further complicated when the health system fails to provide a clear set of guiding rules standardized for all professions, lack of communication with and "buy-in" from all professions involved in the patient's care, and lack of easy access to patient information. Based on these findings in NL, our next steps will be to select strategies for an intervention that can be trialled locally.

Clear endorsement of guidelines by health authorities/hospital administration and development of local policies about who is responsible for ordering tests which are clearly communicated to all surgical HCPs (surgeons, anesthetists, pre-admission clinic staff, and potentially GPs) could address most of the barriers we identified. This includes additional issues stemming from the "just in case" types of ordering which would likely be abated if the direction of the hospital is clear and if HCPs can support their decisions to not order tests using easy to access and up-to-date data from patients' medical records.

In the literature we found only three studies that have tried to reduce preoperative testing, all of which used strategies similar to what we have proposed above (two in the US [49,50] and one in Canada [51]) with the exception of clarifying responsibility for ordering. All interventions included some form of education for HCPs involved in ordering. In addition, two included decision support tools [49,51] and/or use of champions [50,51] to reinforce educational messages. While all studies showed promising results, certainty of the evidence is low. Trials in this area are long overdue. Recently, Ahmadi et al (2021) published a protocol for a

randomized superiority trial planning to implement educational messages triggered by preoperative test ordering the EHR [52]. In Canada, a trial has been registered that aims to reduce preoperative testing for low-risk ambulatory procedures by improving accountability for ordering. They will do this by implementing a hospital policy directing preoperative tests to be ordered only by consulting anesthesiologists based on their clinical assessment of the patient undergoing surgery [53].

## Acknowledgments

We would like to acknowledge Daphne To and Krystal Bursey in their role to help prepare the manuscript for publication and submission. We would like to acknowledge the wider De-implementing Wisely group for their involvement in the identification and prioritization of the research topic as well as their consultation and guidance on choosing the methodological approach for developing behaviour change interventions to reduce low value care.

## Author Contributions

**Conceptualization:** Amanda Hall, Andrea Patey, Kyle Kirkham, Jeremy Grimshaw.

**Formal analysis:** Andrea Pike, Andrea Patey, Sameh Mortazhejri, Samantha Inwood.

**Funding acquisition:** Amanda Hall, Andrea Patey, Kyle Kirkham, Krista Mahoney, Jeremy Grimshaw.

**Investigation:** Amanda Hall, Andrea Pike.

**Methodology:** Amanda Hall, Andrea Pike, Andrea Patey, Sameh Mortazhejri, Samantha Inwood, Shannon Ruzycki, Kyle Kirkham, Krista Mahoney, Jeremy Grimshaw.

**Project administration:** Andrea Pike, Krista Mahoney, Jeremy Grimshaw.

**Supervision:** Amanda Hall, Jeremy Grimshaw.

**Validation:** Andrea Pike, Kyle Kirkham, Jeremy Grimshaw.

**Writing – original draft:** Amanda Hall, Andrea Pike.

**Writing – review & editing:** Amanda Hall, Andrea Pike, Andrea Patey, Sameh Mortazhejri, Samantha Inwood, Shannon Ruzycki, Kyle Kirkham, Krista Mahoney, Jeremy Grimshaw.

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
