## [Decision Letter · Decision Letter 0]

21 Nov 2022

Barriers to reducing preoperative testing for low-risk surgical procedures: A qualitative assessment guided by the Theoretical Domains Framework

PONE-D-22-20721

Dear Dr. Hall,

We’re pleased to inform you that your manuscript has been judged scientifically suitable for publication and will be formally accepted for publication once it meets all outstanding technical requirements.

Kind regards,

Apurva Kumar Pandya, PhD

Academic Editor

PLOS ONE

1.Thank you for stating the following financial disclosure:

“1. JG, AH, KK, SR, APa, KM

2. JG, AH, KK, SR, APa, KM - De-implementing low value care: A research program of the Choosing Wisely Canada Implementation Research Network - grant number 398 527.

3. Canadian Institute of Health Research (CIHR)

4. https://cihr-irsc.gc.ca/e/193.html

5. No”

Please respond by return e-mail so that we can amend your financial disclosure and competing interests on your behalf.

2. Please respond by return e-mail so that we can expand the acronym “JG, AH, KK, SR, APa, KM” in your financial disclosure so that it states the name of your funders in full. 

We will amend your financial disclosure and competing interests on your behalf.

Additional Editor Comments:

I congratulate authors for very good work. Manuscript is well written. It is accepted for publication.

Reviewers' comments:

Reviewer's Responses to Questions

**Comments to the Author**

1. Is the manuscript technically sound, and do the data support the conclusions?

Reviewer #1: Yes

2. Has the statistical analysis been performed appropriately and rigorously? 

Reviewer #1: N/A

3. Have the authors made all data underlying the findings in their manuscript fully available?

Reviewer #1: Yes

4. Is the manuscript presented in an intelligible fashion and written in standard English?

Reviewer #1: Yes

5. Review Comments to the Author

Reviewer #1: The submission is an important contribution to the current perioperative literature. Only a few studies in this area have been conducted and it is important to demonstrate that the findings are reliable in various settings. Newfoundland represents a remote area of Canada which may have different drivers than a more urban population that were represented in the other studies.

The methodology was well described and offers an important narrative in this complex area for future study of the barriers and enablers.

The manner in which the results were presented through shared quotes was well done and very "real".

This paper is an important contribution to our real world work and it will help communicate our current shortcomings in preoperative testing that many leaders and HCPs will understand.

There are not concerns or revisions suggested for this well written paper.

6. PLOS authors have the option to publish the peer review history of their article (what does this mean?). If published, this will include your full peer review and any attached files.

Reviewer #1: **Yes: **Sukhbir S. Singh

---

## [Editor Report · Acceptance letter]

29 Nov 2022

PONE-D-22-20721 

Barriers to reducing preoperative testing for low-risk surgical procedures: A qualitative assessment guided by the Theoretical Domains Framework 

Dear Dr. Hall:

I'm pleased to inform you that your manuscript has been deemed suitable for publication in PLOS ONE. Congratulations! Your manuscript is now with our production department. 

Kind regards, 

on behalf of

Dr. Apurva Kumar Pandya 

Academic Editor

PLOS ONE